# IMPROVE MATHEMATICAL REASONING IN LANGUAGE MODELS WITH AUTOMATED PROCESS SUPERVISION

## ABSTRACT

Complex multi-step reasoning tasks, such as solving mathematical problems or generating code, remain a significant hurdle for even the most advanced large language models (LLMs). Verifying LLM outputs with an Outcome Reward Model (ORM) is a standard inference-time technique aimed at enhancing the reasoning performance of LLMs. However, this still proves insufficient for reasoning tasks with a lengthy or multi-hop reasoning chain, where the intermediate outcomes are neither properly rewarded nor penalized. Process supervision addresses this limitation by assigning intermediate rewards during the reasoning process. To date, the methods used to collect process supervision data have relied on either human annotation or per-step Monte Carlo estimation, both prohibitively expensive to scale, thus hindering the broad application of this technique. In response to this challenge, we propose a novel divide-and-conquer style Monte Carlo Tree Search (MCTS) algorithm named *OmegaPRM* for the efficient collection of high-quality process supervision data. This algorithm swiftly identifies the first error in the Chain of Thought (CoT) with binary search and balances the positive and negative examples, thereby ensuring both efficiency and quality. As a result, we are able to collect over 1.5 million process supervision annotations to train Process Reward Models (PRMs). This fully automated process supervision alongside the weighted self-consistency algorithm is able to enhance LLMs' math reasoning performances. We improved the success rates of the instruction-tuned Gemini Pro model from 51% to 69.4% on MATH500 and from 86.4% to 93.6% on GSM8K. Similarly, we boosted the success rates of Gemma2 27B from 42.3% to 58.2% on MATH500 and from 74.0% to 92.2% on GSM8K. The entire process operates without any human intervention or supervision, making our method both financially and computationally cost-effective compared to existing methods.

## 1 INTRODUCTION

Despite the impressive advancements achieved by scaling Large Language Models (LLMs) on established benchmarks (Wei et al., 2022a), cultivating more sophisticated reasoning capabilities, particularly in domains like mathematical problem-solving and code generation, remains an active research area. Chain-of-thought (CoT) generation is crucial for these reasoning tasks, as it decomposes complex problems into intermediate steps, mirroring human reasoning processes. Prompting LLMs with CoT examples (Wei et al., 2022b) and fine-tuning them on question-CoT solution pairs (Perez et al., 2021; Ouyang et al., 2022) have proven effective, with the latter demonstrating superior performance. Furthermore, the advent of Reinforcement Learning with Human Feedback (RLHF; Ouyang et al., 2022) has enabled the alignment of LLM behaviors with human preferences through reward models, significantly enhancing model capabilities.

Beyond prompting and further training, developing effective decoding strategies is another crucial avenue for improvement. Self-consistency decoding (Wang et al., 2023) leverages multiple reasoning paths to arrive at a voted answer. Incorporating a verifier, such as an off-the-shelf LLM (Huang et al., 2022; Luo et al., 2023), can further guide LLMs in reasoning tasks by providing a feedback loop to verify final answers, identify errors, and suggest corrections. However, the gain of such approaches remains limited for complex multi-step reasoning problems. Reward models offer a promising alternative to verifiers, enabling the reranking of candidate outcomes based on reward signals to ensure higher accuracy. Two primary types of reward models have emerged: Outcome

Reward Models (ORMs; Yu et al., 2024; Cobbe et al., 2021), which provide feedback only at the end of the problem-solving process, and Process Reward Models (PRMs; Li et al., 2023; Uesato et al., 2022; Lightman et al., 2023), which offer granular feedback at each reasoning step. PRMs have demonstrated superior effectiveness for complex reasoning tasks by providing such fine-grained supervision.

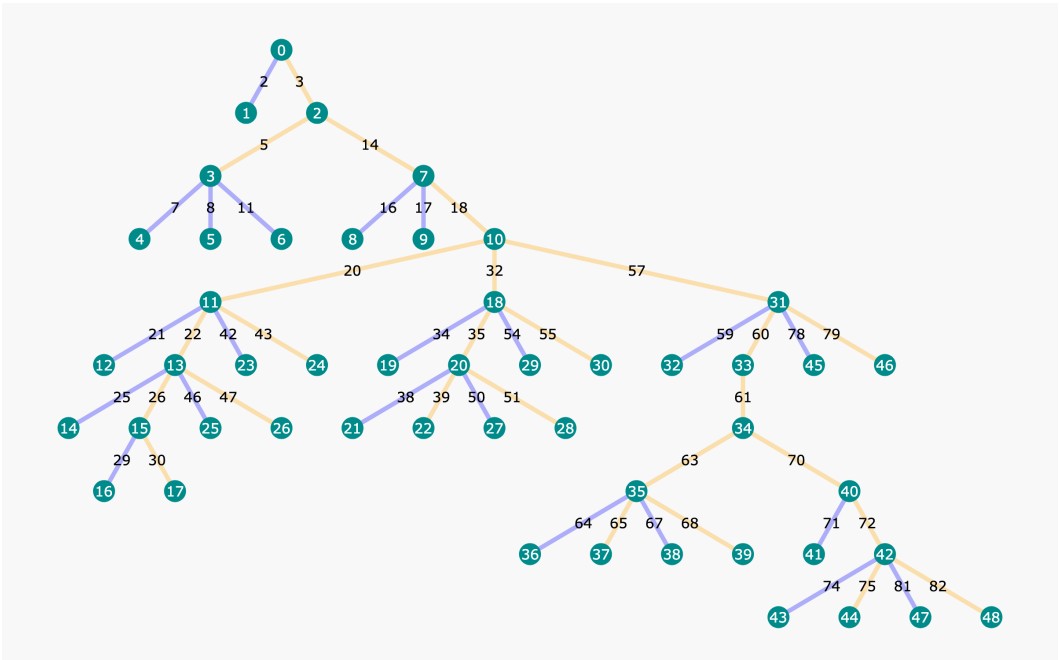

Figure 1: Example tree structure built with our proposed OmegaPRM algorithm. Each node in the tree indicates a state of partial chain-of-thought solution, with information including accuracy of rollouts and other statistics. Each edge indicates an action, *i.e.*, a reasoning step, from the last state. Yellow edges are correct steps and blue edges are wrong.

The primary bottleneck in developing PRMs lies in obtaining process supervision signals, which require supervised labels for each reasoning step. Current approaches rely heavily on costly and labor-intensive human annotation (Uesato et al., 2022; Lightman et al., 2023). Automating this process is crucial for scalability and efficiency. While recent efforts using per-step Monte Carlo estimation have shown promise (Wang et al., 2024a;b), their efficiency remains limited due to the vast search space. To address this challenge, we introduce OmegaPRM, a novel divide-and-conquer Monte Carlo Tree Search (MCTS) algorithm inspired by AlphaGo Zero (Silver et al., 2017) for automated process supervision data collection. For each question, we build a Monte Carlo Tree, as shown in Fig. 1, with the details explained in §3.3. This algorithm enables efficient collection of over 1.5 million high-quality process annotations without human intervention. Our PRM, trained on this dataset and combined with weighted self-consistency decoding, significantly improves the performance of instruction-tuned Gemini Pro from 51% to 69.4% on MATH500 (Lightman et al., 2023) and from 86.4% to 93.6% on GSM8K (Cobbe et al., 2021). We also boosted the success rates of Gemma2 27B from 42.3% to 58.2% on MATH500 and from 74.0% to 92.2% on GSM8K.

Our main contributions are as follows:

- We propose a novel divide-and-conquer style Monte Carlo Tree Search algorithm for automated process supervision data generation.
- The algorithm enables the efficient generation of over 1.5 million process supervision annotations, representing the largest and highest quality dataset of its kind to date. Additionally, the entire process operates without any human annotation, making our method both financially and computationally cost-effective.
- We combine our verifier with weighted self-consistency to further boost the performance of LLM reasoning. We significantly improves the success rates from 51% to 69.4% on

MATH500 and from 86.4% to 93.6% on GSM8K for instruction-tuned Gemini Pro. For Gemma2 27B, we also improved the success rates of from 42.3% to 58.2% on MATH500 and from 74.0% to 92.2% on GSM8K.

## 2 RELATED WORK

**Improving mathematical reasoning ability of LLMs.** Mathematical reasoning poses significant challenges for LLMs, and it is one of the key tasks for evaluating the reasoning ability of LLMs. With a huge amount of math problems in pretraining datasets, the pretrained LLMs (OpenAI, 2023; Gemini Team et al., 2024; Touvron et al., 2023) are able to solve simple problems, yet struggle with more complicated reasoning. To overcome that, the chain-of-thought (Wei et al., 2022b; Fu et al., 2023) type prompting algorithms were proposed. These techniques were effective in improving the performance of LLMs on reasoning tasks without modifying the model parameters. The performance was further improved by supervised fine-tuning (SFT; Cobbe et al., 2021; Liu et al., 2024; Yu et al., 2023) with high quality question-response pairs with full CoT reasoning steps.

**Application of reward models in mathematical reasoning of LLMs.** To further improve the LLM's math reasoning performance, verifiers can help to rank and select the best answer when multiple rollouts are available. Several works (Huang et al., 2022; Luo et al., 2023) have shown that using LLM as verifier is not suitable for math reasoning. For trained verifiers, two types of reward models are commonly used: Outcome Reward Model (ORM) and Process Reward Model (PRM). Both have shown performance boost on math reasoning over self-consistency (Cobbe et al., 2021; Uesato et al., 2022; Lightman et al., 2023), yet evidence has shown that PRM outperforms ORM (Lightman et al., 2023; Wang et al., 2024a). Generating high quality process supervision data is the key for training PRM, besides expensive human annotation (Lightman et al., 2023), Math-Shepherd (Wang et al., 2024a) and MiPS (Wang et al., 2024b) explored Monte Carlo estimation to automate the data collection process with human involvement, and both observed large performance gain. Our work shared the essence with MiPS and Math-Shepherd, but we explore further in collecting the process data using MCTS.

**Monte Carlo Tree Search (MCTS).** MCTS (Świechowski et al., 2021) has been widely adopted in reinforcement learning (RL). AlphaGo (Silver et al., 2016) and AlphaGo Zero (Silver et al., 2017) were able to achieve great performance with MCTS and deep reinforcement learning. For LLMs, there are planning algorithms that fall in the category of tree search, such as Tree-of-Thought (Yao et al., 2023) and Reasoning-via-Planing (Hao et al., 2023). Recently, utilizing tree-like decoding to find the best output during the inference-time has become a hot topic to explore as well, multiple works (Feng et al., 2023; Ma et al., 2023; Zhang et al., 2024; Tian et al., 2024; Feng et al., 2024; Kang et al., 2024) have observed improvements in reasoning tasks.

## 3 METHODS

### 3.1 PROCESS SUPERVISION

Process supervision is a concept proposed to differentiate from outcome supervision. The reward models trained with these objectives are termed Process Reward Models (PRMs) and Outcome Reward Models (ORMs), respectively. In the ORM framework, given a query $q$ (*e.g.*, a mathematical problem) and its corresponding response $x$ (*e.g.*, a model-generated solution), an ORM is trained to predict the correctness of the final answer within the response. Formally, an ORM takes $q$ and $x$ and outputs the probability $p = \mathrm{ORM}(q, x)$ that the final answer in the response is correct. With a training set of question-answer pairs available, an ORM can be trained by sampling outputs from a policy model (*e.g.*, a pretrained or fine-tuned LLM) using the questions and obtaining the correctness labels by comparing these outputs with the golden answers.

In contrast, a PRM is trained to predict the correctness of each intermediate step $x_t$ in the solution. Formally, $p_t = \mathrm{PRM}([q, x_{1:t-1}], x_t)$, where $x_{1:i} = [x_1, \ldots, x_i]$ represents the first $i$ steps in the solution. This provides more precise and fine-grained feedback than ORMs, as it identifies the exact location of errors. Process supervision has also been shown to mitigate incorrect reasoning in the domain of mathematical problem solving. Despite these advantages, obtaining the intermediate signal

for each step's correctness to train such a PRM is non-trivial. Previous work (Lightman et al., 2023) has relied on hiring domain experts to manually annotate the labels, which is and difficult to scale.

## 3.2 PROCESS ANNOTATION WITH MONTE CARLO METHOD

In two closely related works, Math-Shepherd (Wang et al., 2024a) and MiPS (Wang et al., 2024b), the authors propose an automatic annotation approach to obtain process supervision signals using the Monte Carlo method. Specifically, a "completer" policy is established that can take a question $q$ and a prefix solution comprising the first $t$ steps $x_{1:t}$ and output the completion — often referred to as a "rollout" in reinforcement learning — of the subsequent steps until the final answer is reached. As shown in Fig. 2(a), for any step of a solution, the completer policy can be used to randomly sample $k$ rollouts from that step. The final answers of these rollouts are compared to the golden answer, providing $k$ labels of answer correctness corresponding to the $k$ rollouts. Subsequently, the ratio of correct rollouts to total rollouts from the $t$-th step, as represented in Eq. (1), estimates the "correctness level" of the prefix steps up to $t$. Regardless of false positives, $x_{1:t}$ should be considered correct as long as any of the rollouts is correct in the logical reasoning scenario.

$$c_t = \text{MonteCarlo}(q, x_{1:t}) = \frac{\text{num(correct rollouts from } t\text{-th step)}}{\text{num(total rollouts from } t\text{-th step)}} \tag{1}$$

Taking a step forward, a straightforward strategy to annotate the correctness of intermediate steps in a solution is to perform rollouts for every step from the beginning to the end, as done in both Math-Shepherd and MiPS. However, this brute-force approach requires a large number of policy calls. To optimize annotation efficiency, we propose a binary-search-based Monte Carlo estimation.

**Monte Carlo estimation using binary search.** As suggested by Lightman et al. (2023), supervising up to the first incorrect step in a solution is sufficient to train a PRM. Therefore, our objective is locating the first error in an efficient way. We achieve this by repeatedly dividing the solution and performing rollouts. Assuming no false positives or negatives, we start with a solution with potential errors and split it at the midpoint $m$. We then perform rollouts for $s_{1:m}$ with two possible outcomes: (1) $c_m > 0$, indicating that the first half of the solution is correct, as at least one correct answer can be rolled out from $m$-th step, and thus the error is in the second half; (2) $c_m = 0$, indicating the error is very likely in the first half, as none of the rollouts from $m$-th step is correct. This process narrows down the error location to either the first or second half of the solution. As shown in Fig. 2(b), by repeating this process on the erroneous half iteratively until the partial solution is sufficiently small (*i.e.*, short enough to be considered as a single step), we can locate the first error with a time complexity of $O(k \log M)$ rather than $O(kM)$ in the brute-force setting, where $M$ is the total number of steps in the original solution.

## 3.3 MONTE CARLO TREE SEARCH

Although binary search improves the efficiency of locating the first error in a solution, we are still not fully utilizing policy calls as rollouts are simply discarded after stepwise Monte Carlo estimation. In practice, it is necessary to collect multiple PRM training examples (*a.k.a.*, triplets of question, partial solution and correctness label) for a question (Lightman et al., 2023; Wang et al., 2024a). Instead of starting from scratch each time, we can store all rollouts during the process and conduct binary searches from any of these rollouts whenever we need to collect a new example. This approach allows for triplets with the same solution prefix but different completions and error locations. Such reasoning structures can be represented as a tree, as described in previous work like Tree of Thought (Yao et al., 2023).

Formally, consider a *state-action tree* representing detailed reasoning paths for a question, where a state $s$ contains the question and all preceding reasoning steps, and an action $a$ is a potential subsequent step from a specific state. The root state is the question without any reasoning steps: $r_{\text{root}} = q$. The policy can be directly modeled by a language model as $\pi(a|s) = \text{LM}(a|s)$, and the state transition function is simply the concatenation of the preceding steps and the action step, *i.e.*, $s' = \text{Concatenate}(s, a)$.

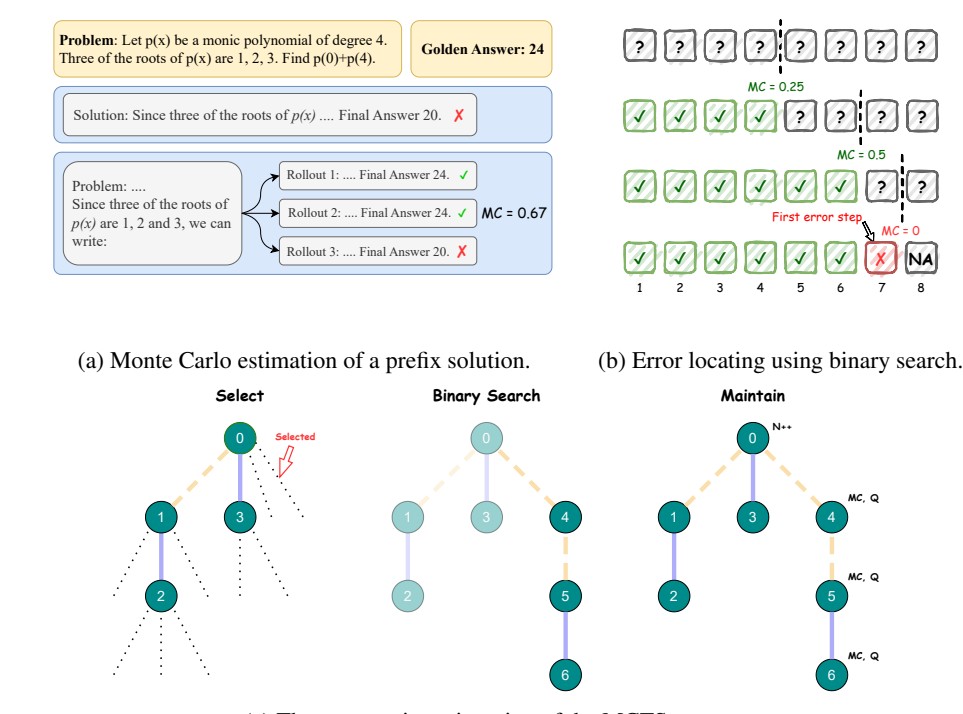

(a) Monte Carlo estimation of a prefix solution.      (b) Error locating using binary search.

(c) Three stages in an iteration of the MCTS process.

Figure 2: Illustration of the process supervision rollouts, Monte Carlo estimation using binary search and the MCTS process. (a) An example of Monte Carlo estimation of a prefix solution. Two out of the three rollouts are correct, producing the Monte Carlo estimation $\mathrm{MC}(q, x_{1:t}) = 2/3 \approx 0.67$. (b) An example of error locating using binary search. The first error step is located at the $7^{\text{th}}$ step after three divide-and-rollouts, where the rollout positions are indicated by the vertical dashed lines. (c) The MCTS process. The dotted lines in Select stage represent the available rollouts for binary search. The bold colored edges represent steps with correctness estimations. The yellow color indicates a correct step, *i.e.*, with a preceding state $s$ that $\mathrm{MC}(s) > 0$ and the blue color indicates an incorrect step, *i.e.*, with $\mathrm{MC}(s) = 0$. The number of dashes in each colored edge indicates the number of steps.

Collecting PRM training examples for a question can now be formulated as constructing such a state-action tree. This reminds us the classic Monte Carlo Tree Search (MCTS) algorithm, which has been successful in many deep reinforcement learning applications (Silver et al., 2016; 2017). However, there are some key differences when using a language model as the policy. First, MCTS typically handles an environment with a finite action space, such as the game of Go, which has fewer than 361 possible actions per state (Silver et al., 2017). In contrast, an LM policy has an infinite action space, as it can generate an unlimited number of distinct actions (sequences of tokens) given a prompt. In practice, we use temperature sampling to generate a fix number of $k$ completions for a prompt, treating the group of $k$ actions as an approximate action space. Second, an LM policy can sample a full rollout until the termination state (*i.e.*, reaching the final answer) without too much overhead than generating a single step, enabling the possibility of binary search. Consequently, we propose an adaptation of the MCTS algorithm named **OmegaPRM**, primarily based on the one introduced in AlphaGo (Silver et al., 2016), but with modifications to better accommodate the scenario of PRM training data collection. We describe the algorithm details as below.

**Tree Structure.** Each node $s$ in the tree contains the question $q$ and prefix solution $x_{1:t}$, together with all previous rollouts $\{(s, r_i)\}_{i=1}^{k}$ from the state. Each edge $(s, a)$ is either a single step or a sequence of consecutive steps from the node $s$. The nodes also store a set of statistics,

$$\{N(s), \mathrm{MC}(s), Q(s, r)\},$$

where $N(s)$ denotes the visit count of a state, $\mathrm{MC}(s)$ represents the Monte Carlo estimation of a state as specified in Eq. (1), and $Q(s, r)$ is a state-rollout value function that is correlated to the chance of selecting a rollout during the selection phase of tree traversal. Specifically,

$$Q(s, r) = \alpha^{1-\mathrm{MC}(s)} \cdot \beta^{\frac{\mathrm{len}(r)}{L}}, \tag{2}$$

where $\alpha, \beta \in (0, 1]$ and $L > 0$ are constant hyperparameters; while $\mathrm{len}(r)$ denotes the length of a rollout in terms of number of tokens. $Q$ is supposed to indicate how likely a rollout will be chosen for each iteration and our goal is to define a heuristic that selects the most valuable rollout to search with. The most straightforward strategy is uniformly choosing rollout candidates generated by the policy in previous rounds; however, this is obviously not an effective way. Lightman et al. (2023) suggests surfacing the *convincing wrong-answer* solutions for annotators during labeling. Inspired by this, we propose to prioritize *supposed-to-be-correct wrong-answer* rollouts during selection. We use the term *supposed-to-be-correct* to refer to the state with a Monte Carlo estimation $\mathrm{MC}(s)$ closed to 1; and use *wrong-answer* to refer that the specific rollout $r$ has a wrong final answer. The rollout contains mistakes made by the policy that should have been avoided given its high $\mathrm{MC}(s)$. We expect a PRM that learns to detect errors in such rollouts will be more useful in correcting the mistakes made by the policy. The first component in Eq. (2), $\alpha^{1-\mathrm{MC}(s)}$, has a larger value as $\mathrm{MC}(s)$ is closer to 1. Additionally, we incorporate a length penalty factor $\beta^{\frac{\mathrm{len}(r)}{L}}$, to penalize excessively long rollouts.

**Select.** The selection phase in our algorithm is simpler than that of AlphaGo (Silver et al., 2016), which involves selecting a sequence of actions from the root to a leaf node, forming a trajectory with multiple states and actions. In contrast, we maintain a pool of all rollouts $\{(s_i, r_j^i)\}$ from previous searches that satisfy $0 < \mathrm{MC}(s_i) < 1$. During each selection, a rollout is popped and selected according to tree statistics, $(s, r) = \arg\max_{(s,r)}[Q(s, r) + U(s)]$, using a variant of the PUCT (Rosin, 2011) algorithm,

$$U(s) = c_{\mathrm{puct}} \frac{\sqrt{\sum_i N(s_i)}}{1 + N(s)}, \tag{3}$$

where $c_{\mathrm{puct}}$ is a constant determining the level of exploration. This strategy initially favors rollouts with low visit counts but gradually shifts preference towards those with high rollout values.

**Binary Search.** We perform a binary search to identify the first error location in the selected rollout, as detailed in §3.2. The rollouts with $0 < \mathrm{MC}(s) < 1$ during the process are added to the selection candidate pool. All divide-and-rollout positions before the first error become new states. For the example in Fig. 2(b), the trajectory $s[q] \to s[q, x_{1:4}] \to s[q, x_{1:6}] \to s[q, x_{1:7}]$ is added to the tree after the binary search. The edges $s[q] \to s[q, x_{1:4}]$ and $s[q, x_{1:4}] \to s[q, x_{1:6}]$ are correct, with $\mathrm{MC}$ values of $0.25$ and $0.5$, respectively; while the edge $s[q, x_{1:6}] \to s[q, x_{1:7}]$ is incorrect with $\mathrm{MC}$ value of $0$.

**Maintain.** After the binary search, the tree statistics $N(s)$, $\mathrm{MC}(s)$, and $Q(s, r)$ are updated. Specifically, $N(s)$ is incremented by 1 for the selected $(s, r)$. Both $\mathrm{MC}(s)$ and $Q(s, r)$ are updated for the new rollouts sampled from the binary search. This phase resembles the *backup* phase in AlphaGo but is simpler, as it does not require recursive updates from the leaf to the root.

**Tree Construction.** By repeating the aboved process, we can construct a state-action tree as the example illustrated in Fig. 1. The construction ends either when the search count reaches a predetermined limit or when no additional rollout candidates are available in the pool.

## 3.4 PRM TRAINING

Each edge $(s, a)$ with a single-step action in the constructed state-action tree can serve as a training example for the PRM. It can be trained using the standard classification loss

$$\mathcal{L}_{\mathrm{pointwise}} = \sum_{i=1}^{N} \hat{y}_i \log y_i + (1 - \hat{y}_i) \log(1 - y_i), \tag{4}$$

where $\hat{y}_i$ represents the correctness label and $y_i = \mathrm{PRM}(s, a)$ is the prediction score of the PRM. Wang et al. (2024b) have used the Monte Carlo estimation as the correctness label, denoted as

$\hat{y} = \text{MC}(s)$. Alternatively, Wang et al. (2024a) have employed a binary labeling approach, where $\hat{y} = \mathbf{1}[\text{MC}(s) > 0]$, assigning $\hat{y} = 1$ for any positive Monte Carlo estimation and $\hat{y} = 0$ otherwise. We refer the former option as *pointwise soft* label and the latter as *pointwise hard* label. In addition, considering there are many cases where a common solution prefix has multiple single-step actions, we can also minimize the cross-entropy loss between the PRM predictions and the normalized pairwise preferences following the Bradley-Terry model (Christiano et al., 2017). We refer this training method as *pairwise* approach, and the detailed pairwise loss formula can be found in Section Appendix B.

We use the pointwise soft label when evaluating the main results in §4.1, and a comparion of the three objectives are discussed in §4.3.

## 4 EXPERIMENTS

**Data Generation.** We conduct our experiments on the challenging MATH dataset (Hendrycks et al., 2021). We use the same training and testing split as described in Lightman et al. (2023), which consists of 12K training examples and a subset with 500 holdout representative problems from the original 5K testing examples introduced in Hendrycks et al. (2021). We observe similar policy performance on the full test set and the subset. For creating the process annotation data, we use the questions from the training split and set the search limit to 100 per question, resulting 1.5M per-step process supervision annotations. To reduce the false positive and false negative noise, we filtered out questions that are either too hard or too easy for the model. Please refer to Appendix A for details. We use $\alpha = 0.5$, $\beta = 0.9$ and $L = 500$ for calculating $Q(s, r)$ in Eq. (2); and $c_{\text{puct}} = 0.125$ in Eq. (3). We sample $k = 8$ rollouts for each Monte Carlo estimation.

**Models.** In previous studies (Lightman et al., 2023; Wang et al., 2024a;b), both proprietary models such as GPT-4 (OpenAI, 2023) and open-source models such as Llama2 (Touvron et al., 2023) were explored. In our study, we perform experiments with both proprietary Gemini Pro (Gemini Team et al., 2024) and open-source Gemma2 (Gemma Team et al., 2024) models. For Gemini Pro, we follow Lightman et al. (2023); Wang et al. (2024a) to initially fine-tune it on math instruction data, achieving an accuracy of approximately 51% on the MATH test set. The instruction-tuned model is then used for solution sampling. For open-source models, to maximize reproducibility, we directly use the pretrained Gemma2 27B checkpoint with the 4-shot prompt introduced in Gemini Team et al. (2024). The reward models are all trained from the pretrained checkpoints.

**Metrics and baselines.** We evaluate the PRM-based majority voting results on GSM8K (Cobbe et al., 2021) and MATH500 (Lightman et al., 2023) using PRMs trained on different process supervision data. We choose the product of scores across all steps as the final solution score following Lightman et al. (2023), where the performance difference between product and minimum of scores was compared and the study showed the difference is minor. Baseline process supervision data include PRM800K (Lightman et al., 2023) and Math-Shepherd (Wang et al., 2024a), both publicly available. Additionally, we generate a process annotation dataset with our Gemini policy model using the brute-force approach described in Wang et al. (2024a;b), referred to as Math-Shepherd (our impl) in subsequent sections.

### 4.1 MAIN RESULTS

Table 1 and Fig. 3 presents the performance comparison of PRMs trained on various process annotation datasets. OmegaPRM consistently outperforms the other process supervision datasets. Specifically, the fine-tuned Gemini Pro achieves 69.4% and 93.6% accuracy on MATH500 and GSM8K, respectively, using OmegaPRM-weighted majority voting. For the pretrained Gemma2 27B, it also performs the best with 58.2% and 92.2% accuracy on MATH500 and GSM8K, respectively. It shows superior performance comparing to both human annotated PRM800K but also automatic annotated Math-Shepherd. More specifically, when the number of samples is small, almost all the PRM models outperforme the majority vote. However, as the number of samples increases, the performance of other PRMs gradually converges to the same level of the majority vote. In contrast, our PRM model continues to demonstrate a clear margin of accuracy.

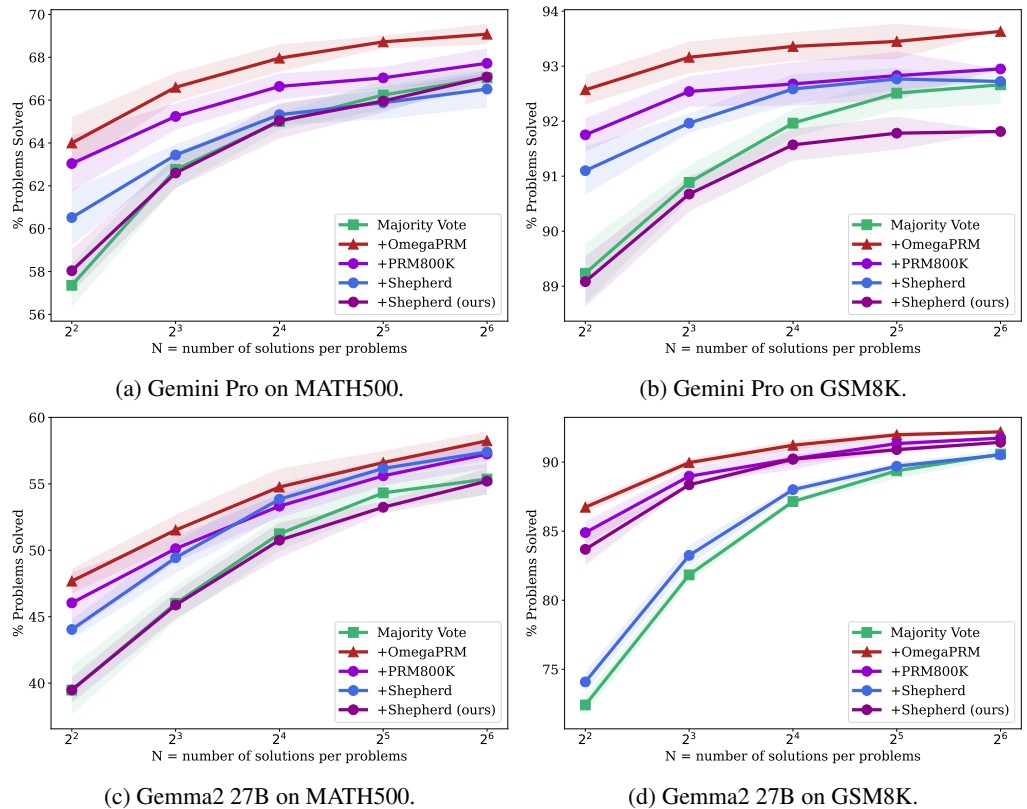

(a) Gemini Pro on MATH500.

(b) Gemini Pro on GSM8K.

(c) Gemma2 27B on MATH500.

(d) Gemma2 27B on GSM8K.

Figure 3: A comparison of PRMs trained with different process supervision datasets, evaluated by their ability to search over many test solutions using a PRM-weighted majority voting. We visualize the variance across many sub-samples of the 128 solutions we generated in total per problem.

Table 1: The performance comparison of PRMs trained with different process supervision datasets. The numbers represent the percentage of problems solved using PRM-weighted majority voting with $k = 64$.

|  | MATH500 | | GSM8K | |
|---|---|---|---|---|
|  | Gemini Pro | Gemma 2 27B | Gemini Pro | Gemma 2 27B |
| MajorityVote@64 | 67.2 | 54.7 | 92.7 | 90.6 |
| + Math-Shepherd | 67.2 | 57.4 | 92.7 | 90.5 |
| + Math-Shepherd (our impl) | 67.2 | 55.2 | 91.8 | 91.4 |
| + PRM800K | 67.6 | 57.2 | 92.9 | 91.7 |
| + OmegaPRM | **69.4** | **58.2** | **93.6** | **92.2** |

## 4.2 STEP DISTRIBUTION

An important factor in process supervision is the number of steps in a solution and the length of each step. Previous works (Lightman et al., 2023; Wang et al., 2024a;b) use rule-based strategies to split a solution into steps, *e.g.*, using newline as delimiters. In contrast, we propose a more flexible method for step division, treating any sequence of consecutive tokens in a solution as a valid step. We observe that many step divisions in Math-Shepherd lack semantic coherence to some extent. Therefore, we hypothesize that semantically explicit cutting is not necessary for training a PRM.

In practice, we first examine the distribution of the number of steps per solution in PRM800K and Math-Shepherd, as shown in Fig. 4, noting that most solutions have less than 20 steps. During binary search, we aim to divide a full solution into 16 pieces. To calculate the expected step length, we

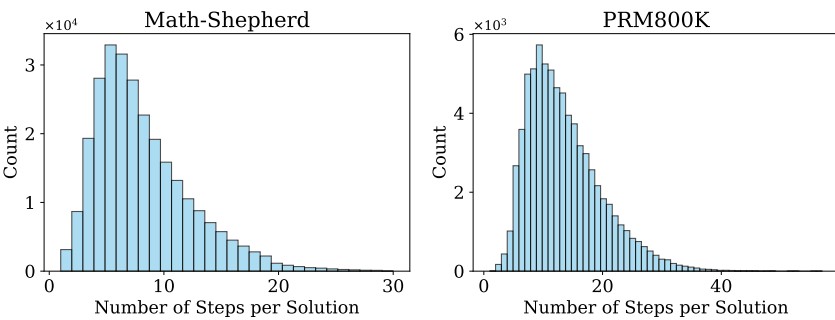

Figure 4: Number of steps distribution.

divide the average solution length by 16. The binary search terminates when a step is shorter than this value. The resulting distributions of step lengths for OmegaPRM and the other two datasets are shown in Fig. 5. This flexible splitting strategy produces a step length distribution similar to that of the rule-based strategy.

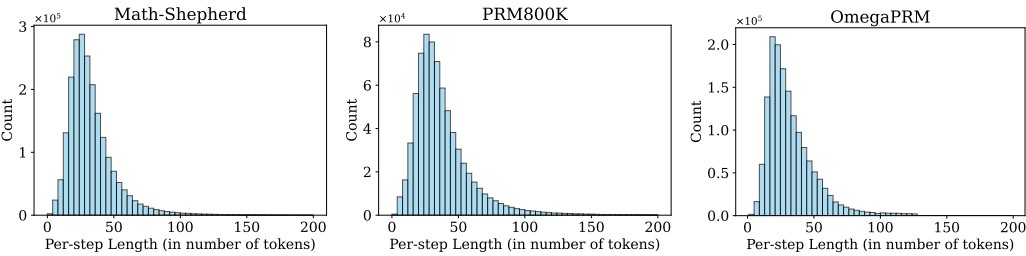

Figure 5: Step length distribution in terms of number of tokens.

### 4.3 PRM Training Objectives

Table 2: Comparison of different training objectives for PRMs.

|                 | Soft Label | Hard Label | Pairwise |
|-----------------|------------|------------|----------|
| PRM Accuracy (%) | **70.1**   | 63.3       | 64.2     |

As outlined in §3.4, PRMs can be trained using multiple objectives. We construct a small process supervision test set using the problems from the MATH test split. We train PRMs using pointwise soft label, pointwise hard label and pairwise loss respectively, and evaluate how accurately they can classify the per-step correctness. Table 2 presents the comparison of different objectives, and the pointwise soft label is the best among them with 70.1% accuracy.

### 4.4 Algorithm Efficiency

As described in Section §3.2 and §3.3, we utilize binary search and Monte Carlo Tree Search to improve the efficiency of OmegaPRM process supervision data collection by effectively identifying the first incorrect step and reusing rollouts in Monte Carlo estimation. To quantitatively measure the efficiency of OmegaPRM, we collected process supervision data using both brute-force-style method (Wang et al., 2024a;b) and OmegaPRM with the same computational budget. As a result, we were able to generate 200K data points using the brute-force algorithm compared to 15 million data points with OmegaPRM, demonstrating a 75-times efficiency improvement. In practice, we randomly down-sampled OmegaPRM data to 1.5 million for PRM training.

## 5 LIMITATIONS

There are some limitations with our paper, which we reserve for future work:

**Automatic process annotation is noisy.** Our method for automatic process supervision annotation introduces noise in the form of false positives and negatives, but experiments indicate that it can still effectively train a PRM. The PRM trained on our dataset performs better than one trained on the human-annotated PRM800K dataset. The precise impact of noise on PRM performance remains uncertain. For future research, a comprehensive comparison of human and automated annotations should be conducted. One other idea is to integrate human and automated annotations, which could result in more robust and efficient process supervision.

**Human supervision is still necessary.** Unlike the work presented in AlphaGo Zero (Silver et al., 2017), our method requires the question and golden answer pair. The question is necessary for LLM to start the MCTS and the golden answer is inevitable for the LLM to compare its rollouts with and determine the correctness of the current step. This will limit the method to the tasks with such question and golden answer pairs. Therefore, we need to adapt the current method further to make it suitable for open-ended tasks.

## 6 CONCLUSION

In conclusion, we introduce OmegaPRM, a divide-and-conquer Monte Carlo Tree Search algorithm, designed to automate the process supervision data collection for LLMs. By efficiently pinpointing the first error in the Chain-of-Thought and balancing data quality, OmegaPRM addresses the shortcomings of existing methods. Our automated approach enables the collection of over 1.5 million process supervision annotations, which are used to train a PRM. Leveraging this automated process supervision with the weighted self-consistency algorithm, we improve LLM mathematical reasoning performance, achieving a 69.4% success rate on the MATH benchmark — a 18.4% absolute increase over the base model which amounts to a relative improvement of 36%. Additionally, our method significantly reduces data collection costs compared to human annotation and brute force Monte-Carlo sampling. These findings highlight OmegaPRM's potential to enhance LLM capabilities in complex multi-step reasoning tasks.

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

APPENDIX

## A    QUESTION FILTERING

During the evaluation of partial solution correctness using MC estimation, false negative noise may be introduced when a question is too hard for the model, thus no correct rollout can be found even with correct partial solution. Or false positive noise may be introduced when a question is too easy, that model can conclude in correct answer given partial solution with wrong step. It is not possible to exclude such noise completely, but we can reduce the chance by filtering out questions that are either too hard or too easy for the model. Specifically, we ran a $k = 32$ rollouts for each question in the 12K training data, and filter out the questions that with no correct answer (too hard) or no wrong answer (too easy) in the 32 rollouts.

## B    PAIRWISE LOSS FORMULA

When training with pairwise labels, the Bradley-Terry model (people typically use this objective to train reward models in RLHF) generally accepts two probability scalars summing up to 1. When we select the two actions as a pair, there are two cases. The first case is that one sample with a zero MC value, and the other sample with a positive MC value. The second case is that both samples are with positive MC values. The first case is straight-forward, and a normalization step is required for the second case.

Assume the two MC values are $p$ and $q$, and they follow the Bernoulli distribution: $P(X = 1) = p$ and $P(Y = 1) = q$. We need to calculate the probability that action X is preferred over action Y and vice versa.

$$
\begin{aligned}
P(X > Y) &= P(X = 1, Y = 0) = p(1 - q), \\
P(X < Y) &= P(X = 0, Y = 1) = (1 - p)q, \\
P(X = Y) &= P(X = 0, Y = 0) + P(X = 1, Y = 1) = (1 - p)(1 - q) + pq.
\end{aligned} \tag{5}
$$

For the tied situation, each action has half the chance of being preferred. Thus,

$$
\begin{aligned}
P(\text{action X is preferred}) &= P(X > Y) + 1/2 * P(X = Y) = 1/2 * (1 + p - q), \\
P(\text{action Y is preferred}) &= P(X < Y) + 1/2 * P(X = Y) = 1/2 * (1 + q - p).
\end{aligned} \tag{6}
$$

Now the MC values are normalized and we can train with the pairwise loss.

