# OpenReview forum: "Improve Mathematical Reasoning in Language Models with Automated Process Supervision"
_ICLR.cc/2025/Conference — Submitted to ICLR 2025_

### Official Review · Reviewer_AhH6 · 2024-10-27

**Soundness:** 3
**Presentation:** 2
**Contribution:** 2
**Rating:** 5
**Confidence:** 4

**Summary:**

This paper proposes an efficient automated method for labeling each step in a process reward model. The core concept involves using binary search to identify the first incorrect step in a rollout and applying a tree-based search to explore the sample space. The resulting artifact contains 1.5 million stepwise annotations. The authors demonstrate the method's effectiveness by comparing the quality of process reward models (PRMs) trained with this dataset to those trained on previously proposed datasets.

**Strengths:**

1. The paper significantly enhances the efficiency of generating process labels by maintaining a tree structure of stepwise derivations and efficiently expanding the tree to minimize the number of required rollouts.
1. The application of binary search to pinpoint the first incorrect step is a novel approach.

**Weaknesses:**

1. Empirical results on the MATH and GSM8K datasets indicate a clear upper bound on this distillation-based self-improvement method, showing that performance improvement is constrained by the initial capability of the base model (e.g., Gemini Pro and Gemma 2).
1. The improvement observed on the GSM8K dataset is particularly questionable, as it shows a less than 2% gain compared to majority voting.
1. This paper needs to improve its presentation. On page 3, L159, the sub-indece in x are inconsistent. On page 4, L163 `which is and difficult _?_ to scale`. On page 5, L267, `(s,r)` stands for a full rollout, while `(s,a)` represents a on-step rollout. The duplicate notations are confusing.

**Questions:**

1. Given the higher inference cost associated with using reward models, does the method ultimately reduce the overall cost of inference?
1. Is the method scalable? For instance, how could one expect to improve MATH500 to 90% accuracy by iterating on this method?
1. How effective is the method when the base model is either very weak or very strong (e.g., performance improvement on o1)?

---

### Official Review · Reviewer_4m15 · 2024-10-30

**Soundness:** 1
**Presentation:** 1
**Contribution:** 2
**Rating:** 3
**Confidence:** 4

**Summary:**

The paper introduces OmegaPRM, a divide-and-conquer Monte Carlo Tree Search (MCTS) algorithm aimed at automating the collection of process supervision data for large language models (LLMs). It efficiently identifies the first error in a reasoning chain, thus improving the performance of LLMs in mathematical reasoning tasks, demonstrated by a significant improvement in accuracy on the MATH benchmark.

**Strengths:**

- The automated process reduces data collection costs significantly compared to traditional human annotation methods.

**Weaknesses:**

- While OmegaPRM's efficiency is emphasized, a direct comparison with methods that rely on human annotations and noise mitigation strategies would enhance its practical evaluation.

- Present only the results of Gemini and Gemma on MATH500 and GSM8K.

- The accuracy gain is minimal and may be due to variance; the accuracy improvement mentioned in the abstract is not solely attributable to OmegaPRM.

- The proposed OmegaPRM primarily combines MCTS and binary search, which limits its novelty.

**Questions:**

- Consider including a broader range of experiments to evaluate the generalizability of OmegaPRM across different reasoning domains.

- It might be helpful to discuss more explicitly the limitations of relying on human-annotated data alongside your automated methods, perhaps by providing quantitative analysis on the impact of noise on performance metrics.

---

### Official Review · Reviewer_26Wf · 2024-11-04

**Soundness:** 3
**Presentation:** 3
**Contribution:** 3
**Rating:** 6
**Confidence:** 3

**Summary:**

This paper proposes to train a PRM as verifier for LLM math reasoning. Since human annotation for step-wise reasoning is expensive, the authors propose to use mcts score as the annotation signals. To enhance the efficiency of construct tree, a binary search method is used to accelerate pruning. Trained on the collected step supervision data, the PRM is able to help LLM outperform previous methods on math reasoning in inference time.

**Strengths:**

- This paper collects a large size of step annotation data without human labelling for math reasoning. Step-wise data is more fine-grained and enable many potentials such as PRM training.

- When preparing the data, the authors contract a space-action tree in math reasoning environment with a efficient binary search approach to locate where a solution starts to make mistake.

- With the trained PRM as verifier, the math reasoning capability improves a bit compared to previous approaches.

**Weaknesses:**

- This paper aims to improve MCTS efficiency by binary search the first incorrect step and reusing rollouts. However, the efficiency gain described in the experiment section is very vague. The authors only report they can generate much more examples with same computational cost. Since efficiency is one major claim of this paper, it would be more clear to give more details, e.g., with same hardware environment, how much exact time the proposed method can save compared to existing approaches.

- A candidate poll of all rollouts from the binary search is maintained for later selection in mcts, which might increases memory / storage overhead. The authors could discuss more, e..g, with experiments results, about the time-space efficiency trade-off.

**Questions:**

- How do the value of $\alpha$.$\beta$, L and $c_{puct}$ are selected empirically?
- Will the collected step annotation data, or the PRM, be released to public?

---

### Meta-Review · Area_Chair_xX7T · 2024-12-23

**Metareview:**

This paper proposes a new MCTS-based algorithm for the efficient collection of high-quality data for learning Process Reward Model (PRM), and show the improvements gained by proposed methods on existing benchmarks for math problems. While reviewers raised several questions concerning the experiment details, efficiency and the legitimacy of results, it's unfortunate that authors was not able to provide any rebuttal to the reviews. We thus recommend rejection.

**Additional Comments On Reviewer Discussion:**

All reviewers raised serious concerns for experiments. There is no author feedback.

---

### Decision · Program_Chairs · 2025-01-22

Reject